# Scene Diffusion: Text-driven Scene Image Synthesis Conditioning on a Single 3D Model

## ABSTRACT

Scene image is one of the important windows for showcasing product design. To obtain it, the standard 3D-based pipeline requires designer to not only create the 3D model of product, but also manually construct the entire scene in software, which hindering its adaptability in situations requiring rapid evaluation. This study aims to realize a novel conditional synthesis method to create the scene image based on a single-model rendering of the desired object and the scene description. In this task, the major challenges are ensuring the strict appearance fidelity of drawn object and the overall visual harmony of synthesized image. The former's achievement relies on maintaining an appropriate condition-output constraint, while the latter necessitates a well-balanced generation process for all regions of image. In this work, we propose Scene Diffusion framework to meet these challenges. Its first progress is introducing the Shading Adaptive Condition Alignment (SACA), which functions as an intensive training objective to promote the appearance consistency between condition and output image without hindering the network's learning to the global shading coherence. Afterwards, a novel low-to-high Frequency Progression Training Schedule (FPTS) is utilized to maintain the visual harmony of entire image by moderating the growth of high-frequency signals in the object area. Extensive qualitative and quantitative results are presented to support the advantages of the proposed method. In addition, we also demonstrate the broader uses of Scene Diffusion, such as its incorporation with ControlNet.

## CCS CONCEPTS

• **Computing methodologies → Computer vision tasks**.

## KEYWORDS

scene image synthesis, conditional image synthesis, text-to-image, diffusion model

## 1 INTRODUCTION

Exhibiting product through scene image is a common practice in modern design. The predominant technique for creating scene images is 3D rendering, which calls for designers to manually construct the entire scene in software. This process not only takes up a significant amount of time, but also necessitates users to possess expert skills in 3D rendering like texturing and lighting. These

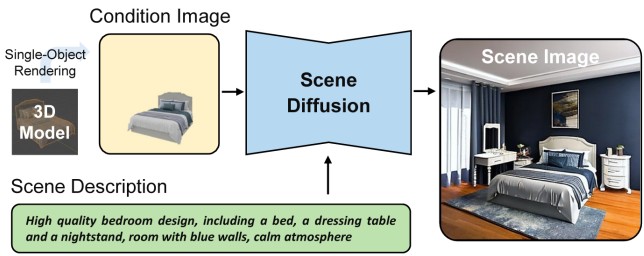

**Figure 1: Scene Diffusion employs the diffusion model principle and generates the scene image conditioning on the rendering of single 3D model and the scene description.**

factors clearly impede its adaptability in situations requiring rapid evaluation. Given the successes of text-to-image(T2I) techniques recently [3, 4, 20, 24, 25, 37], in this work, we attempt to realize a novel conditional synthesis framework to create scene image solely based on a single 3D model and the scene description. As depicted in Figure 1, upon creating the 3D model of designed object, the user only needs to provide a single-model rendering image in the desired position and posture, the network can accordingly generate a scene image that match the scene description. This framework streamlines the laborious scene construction in traditional 3D pipelines, leading to a marked improvement in design efficiency.

The synthesized scene image is expected to meet two major criteria: The first is the **appearance fidelity**. In general, all appearance details of the object contribute to its design. A qualified scene image should faithfully present these minutiae. For this criterion, the direct solution would be to enforce a consistency constraint between the condition and the output image. However, since the condition image is rendered without scene context, the displayed colors of object, known as its shading, is bound to be different from that in the targeted output image. The objects' shading in the latter is coherent with the entire scene. Enforcing the network's prediction to be consistent with both of them is impractical and will undoubtedly disturb the network's learning to the prior of global shading coherence. Therefore, the first challenge we encounter is coordinating the network's learning to the object appearance consistency and the global shading coherence. The second criterion is the **visual harmony** of entire image. In the envisioned diffusion-based framework [5, 11], condition image and description text are responsible to prompt the object area and background area of scene image respectively. Because the former already contains the primary information about the object, it is highly probable that the denoising processes of the two areas will not synchronize, resulting in a disharmony visual effect. In view of this, the second challenge we must overcome is adapting the generation process of the object and background areas to alleviate the issue of visual disharmony.

*ACM MM, 2024, Melbourne, Australia*

© 2024 Copyright held by the owner/author(s). Publication rights licensed to ACM.
ACM ISBN 978-x-xxxx-xxxx-x/YY/MM
https://doi.org/10.1145/nnnnnnn.nnnnnnn

In this work, we propose Scene Diffusion to meet the challenges. The first novelty of it is introducing the Shading Adaptive Condition Alignment (SACA) as an intensive training objective. In SACA, the shading difference between the condition and targeted output image is described though a parametric shading adaptive transformation. The shading of the condition image will be adapted to resemble that of the targeted output image through this transformation. Then the pixel-level error between converted condition and current output is optimized to achieve alignment. SACA ensures a reliable consistency constraint between the condition and output image without impeding the network's learning to the prior of global shading coherence. It assists the network in successfully achieving the goal of maintaining appearance fidelity. In terms of visual harmony, we attribute the observed disharmony issue to the abnormal growth of high-frequency signals in the object area, and develop a Frequency Progression Training Schedule (FPTS) to tackle the problem. When the timestep is large, the network mainly learns low-frequency information, and the SACA is only executed between the low-frequency bands of the condition and output image. With timestep decreasing, the constraint gradually spreads to the higher frequencies. The quantitative analysis demonstrates that FPTS effectively slows down the growth of high-frequency signals in the object area, ultimately enhancing the visual harmony of the entire image.

The main contributions of this article are as follows:

- Investigating a novel challenge of generating scene images based on a single 3D model with the prior of the pre-trained T2I model. In contrast to conventional 3D-based pipeline, this method streamlines the complex scene crafting process and is better suited for the scenarios that demand rapid evaluation.
- Proposing a novel learning-based scene image synthesis framework. By introducing Shading Adaptive Condition Alignment as an intensive training objective, the framework effectively facilitates the network's learning to the complex relationship between the condition and output image. The subsequent addition of Frequency Progression Training Schedule further improves the visual harmony of the synthesized image.
- The proposed Scene Diffusion has been shown to excel beyond the other alternatives in terms of faithfully presenting condition object and generating high quality images. In addition,it also shows the ability to seamlessly integrate with existing ControlNet and potential for generalizing to real image fragments.

## 2 RELATED WORKS

Recently, the emergence of large-scale T2I models [3, 4, 20, 24, 25, 37] has significantly advanced the progress of image synthesis. Based on that, numerous new image-editing [2, 38, 40] and conditional generation techniques [12, 19, 39, 41] were developed and have shown significant success in practice. In this paper, we try to accomplish a fresh task that synthesizing the scene image conditioning on the single-model rendering and the description text. This task necessitates the model to not only produce a suitable background, but also accurately present the details of the object. To

our best knowledge, the technologies with the potential to complete this task can be segmented into three categories: personalization-based method, editing-based method, and learning-based method. In this section, we will provide a brief overview of these related studies and analyze their applicability in our task.

**Personalization-based Method** Personalization techniques [26, 27, 31] enable the model to learn a specific object from the limited references and then generate images about it. The early methods such as DreamBooth [26] typically involved a brief tuning process. Nowadays, the focus has shifted to the tuning-free personalization [8, 13, 15, 32, 34] among researches. A representative method was BLIP-Diffusion [15]. In this method, a cross-modal encoder was employed to embed the identity of object from reference image. The output embedding would then be combined with the text embedding to jointly prompt the image synthesis process. By integrating with edge-driven ControlNet, BLIP-Diffusion could further achieve control over the object's position and posture, which theoretically met the objectives of our task. However, since condition image contained only a single object, the edge map obtained from it could hardly guide the generation of background area. As shown in the experiment section, the images generated by this scheme generally had the relatively monotonous background.

**Editing-based Method** Since scene image synthesis can be considered a specific image editing task, our comparison also includes editing-based methods [2, 10, 25, 35, 38, 40]. In this field, a commonly applied technique was SD-inpainting [25]. This method effectively utilized the prior knowledge of the pre-trained diffusion-based T2I model regarding image composition. In denoising process, a mask was applied to fix the object area, while the rest was synthesized from random noise in accordance with the text prompts. This strategy guaranteed the preservation of the object area's information, but it also meant that the object's shading would not be altered to match the surroundings. The Prompt2Prompt [10] provided a more flexible editing paradigm by manipulating the features of cross-attention layers, by which users could modify the semantics of any elements in the image, including the background. Following researches like InstructPix2Pix [2] utilized Prompt2Prompt to generate paired image editing datasets and then trained an instruction-driven image editing model, ultimately enhancing the user-friendliness of this type of methods. An aspect of these methods was that the editing operation was reliant on the original image's structure. In our task, the condition image was comprised solely of foreground object. It would be challenging for these methods to produce a scene image with extensive background context without the reference.

**Learning-based Method** In existing T2I models, text was mainly utilized to prompt the semantic information of image. For controlling other elements like image structure, specific conditional image generation methods [12, 19, 22, 39, 41] were frequently employed. The most influential technology is ControlNet [39]. This method employed the Stable Diffusion [25] as the base model and utilized paired condition-output data to train the additional condition branch. Due to the rich image priors of large-scale T2I model, ControlNet was able to learn the complex tasks such as depth-to-image and edge-to-image with relatively limited data. Additionally, some studies have made headway in tasks such as layout-to-image [14, 36, 42], , pose-to-image [18, 28] and sketch-to-image [33]. A

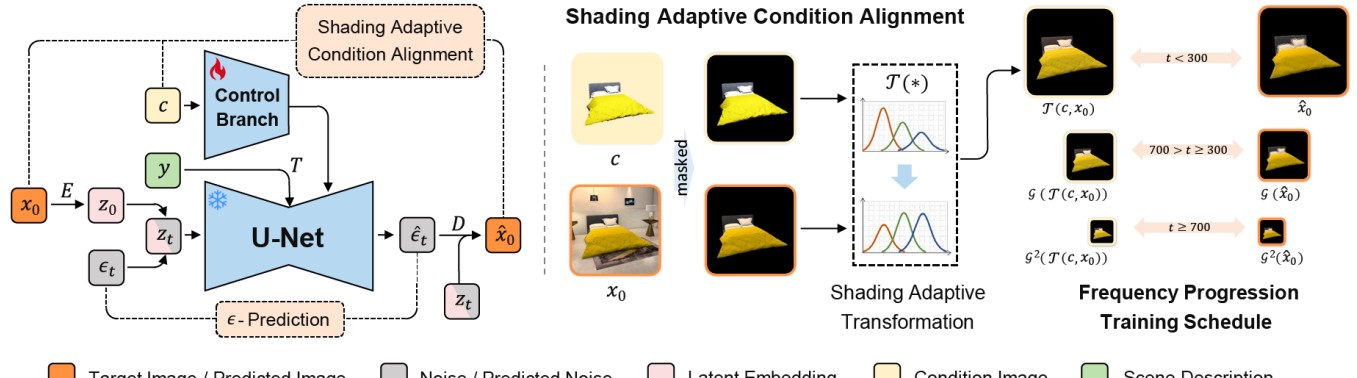

Figure 2: The *left* diagram gives the overview of Scene Diffusion framework. Refer to the legends below for the meanings of blocks. The *right* diagram illustrates the principles of proposed Shading Adaptive Condition Alignment training objective and Frequency Progression Training Schedule.

noteworthy fact was that these methods generally did not introduce explicit condition-output constraints during the training process. This strategy yielded satisfactory results in the above tasks that did not require strict control over appearance. However, in our task, the synthesized scene image was required to faithfully present the appearance details of condition object. The network was unable to fulfill this requirement by autonomously learning the relationship between the condition and output image. In the experiment section, the effects of the proposed method and original ControlNet will be compared to verify the necessity of this constraint.

## 3 METHODOLOGY

In this section, we first provide an overview of Scene Diffusion, followed by detailed introduction of two key improvements.

### 3.1 Overview

The goal of Scene Diffusion framework is to learn a conditional distribution $p(x_0|y, c)$. Given a single-object rendering image $c \in \mathbb{R}^{h \times w \times ch}$ and a scene description text $y$, model is expected to generate a high quality scene image $x \in \mathbb{R}^{h \times w \times ch}$. While ensuring that the scene semantics conforms to the text description, $x$ is expected to accurately present the position, posture, and appearance details of the condition object. Our approach adopts the diffusion-based generative model and utilizes the identical network architecture as ControlNet [39]. As shown in Figure 2, the network mainly consists of two parts. The first is the U-Net inherited directly from Stable Diffusion. Also taken are Stable Diffusion's image encoder $E$, image decoder $D$ and text encoder $T$. The second is condition branch, which is constructed with the duplication of U-Net's encoder fragment and a randomly initialized input block. Condition branch takes $c$ as input. The feature maps output by its blocks are incorporated into the features of the corresponding blocks of U-Net after the zero-convolution layer. Training only affects the parameters of the condition branch, leaving all the other parts frozen.

In training phase, the target scene image will be initially encoded as the latent embedding $z_0 = E(x_0)$. The purpose of this operation is to reduce the amount of computation of U-Net. Then $z_0$ will be fused with the random noise $\epsilon$. U-Net is trained to predict the noise component from the noisy input. This training objective is called $\epsilon - prediction$, expressed as:

$$\mathcal{L}_{\epsilon-pred} = \mathbb{E}_{(z_0, y, c), t}[||\epsilon_t - \Phi(z_t, y, c, t)||_2^2] \quad (1)$$

$t \in \{1, \ldots 1000\}$ represents the timestep of diffusion process. $z_t$ is the noisy intermediate latent embedding, calculated as $z_t = \sqrt{\overline{\alpha}_t}z_0 + \sqrt{1 - \overline{\alpha}_t}\epsilon_t$, in which $\overline{\alpha}_t$ is the predefined diffusion scheduling parameter. $\epsilon_t$ is the random noise added at the current timestep. $\Phi$ is the trained network to predict $\epsilon_t$. For each timestep, we can get a prediction for the target image $\hat{x}_0 = D(\sqrt{1/\overline{\alpha}_t}z_t - \sqrt{(1 - \overline{\alpha}_t)/\overline{\alpha}_t}\hat{\epsilon}_t)$, $\hat{\epsilon}_t = \Phi(z_t, y, c, t)$. Since $\overline{\alpha}_t$ and the image decoder $D$ are deterministic, the training objective of $\epsilon - prediction$ is theoretically equivalent to directly predict $x_0$.

The above are the basic components of proposed method. To better meet the challenges of scene image synthesis, two improvements are further applied. The first is introducing Shading Adaptive Condition Alignment (SACA) as the intensive training objective. Its function is to promote the appearance fidelity between condition and output image without hindering the network's learning to prior of the global shading coherence. The inclusion of Frequency Progression Training Schedule (FPTS) afterwards further improves the visual harmony by moderating the growth of high-frequency signals in the object area. Subsequently, we will provide a thorough introduction to them and demonstrate their effectiveness through theoretical analysis and quantitative evaluation.

### 3.2 Shading Adaptive Condition Alignment

It has been demonstrated that the constraint between the condition and output image is crucial for ensuring the appearance fidelity of synthesized image. The following concern is which form it should adopt. The condition image $c$ is rendered without scene context. Its shading is bound to differ from the targeted scene image in which the object's shading has strong coherence with the overall environment. As stated above, the optimization goal of $\mathcal{L}_{\epsilon-pred}$ is equivalent to predict $x_0$ in theory. Directly enforcing the RGB color

of the predicted image $\hat{x}_0$ to conform to $c$ not only theoretically goes against $\mathcal{L}_{\epsilon-pred}$, but also impedes the network's learning to the prior of global shading coherence in practice.

In this scenario, the logical option would be to first adapt $c$'s shading to be same as $x_0$ via a transformation $\mathcal{T}$ and then conduct consistency constraint between $\hat{x}_0$ and converted condition image, expressed as:

$$\mathcal{L}_{saca} = \mathbb{E}_{(x_0,y,c),t}[||m \odot (\hat{x}_0 - \mathcal{T}(c, x_0)||_2^2] \tag{2}$$

$m$ is mask of object area. The transformation $\mathcal{T}$ needs to satisfy two conditions: first, it must be parameterized to allow for gradient propagation, and second, it should be able to convert $c$ to the same shading as $x_0$ with maximum precision. Finally, we consider the following form:

$$\mathcal{T}(c, x_0) = std(x_0) * \frac{c - mean(c)}{std(c)} + mean(x_0) \tag{3}$$

The transformation is preformed separately on each channel in the object area. For simplicity we omit $m$ in the formulation. The mathematical meaning of $\mathcal{T}$ is to adjust the color distribution of $c$ to be consistent with $x_0$. Subsequently, we will provide a concise analysis to validate the suitability of $\mathcal{T}$ in adapting the changes in ambient light color of object.

Here we employ the fundamental Blinn-Phong [1, 21] shading model to assist in our analysis. According to it, the displayed color $l$ of a point on the object surface can be expressed as:

$$l = l_a + l_d + l_s \tag{4}$$

$l_a, l_d$ and $l_s$ are ambient lighting, diffuse reflection and specular highlight colors respectively. Generally, $l_a$ is regarded as the main contributor to displayed color, which calculated as:

$$l_a = K_a I_a \tag{5}$$

$K_a$ is the ambient light absorption rate of this point, which is associated with the its intrinsic color and material. $I_a$ is the ambient light intensity. Since ambient light involves the reflected light from all possible directions, $l_a$ is usually defined as a constant across the whole surface.

Supposing $K_a$ follows a normal distribution. For ambient light colors of the condition image and the targeted output image, $c_a$ and $(x_0)_a$, we have:

$$\frac{(x_0)_a - mean((x_0)_a)}{std((x_0)_a)} \equiv \frac{c_a - mean(c_a)}{std(c_a)} \tag{6}$$

Through basic manipulation, we obtain the transformation in the Equation 3. $l_d$ and $l_s$ are influenced by the light sources and are not addressed in this work. Experiments show that the current $\mathcal{T}$ can address the primary color differences between $c$ and $x_0$. In supplementary materials, intuitive results are provided to demonstrate its effects.

However, due to the simplifications implemented, there will inevitably be errors between $\mathcal{T}(c, x_0)$ and original $x_0$. To avoid the affects of the remaining error on the optimization of $\mathcal{L}_{\epsilon-pred}$, we formulated the following strategy: for every pixel in object

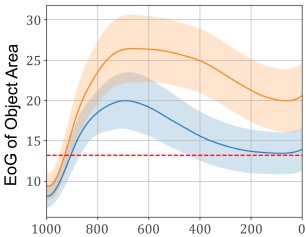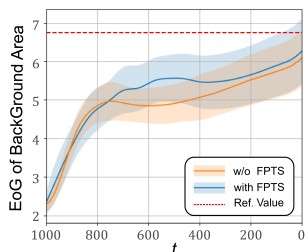

**Figure 3: The changing trends of the EoG of object and background areas during the diffusion sampling process. The solid lines reflect the average values, while the bounds of filled areas reflect the $\pm 1/2$ standard deviations. The results are calculated based on 50 test samples. The reference value is calculated on the corresponding target images.**

area, when $\hat{x}_{(i,j)} \notin [x_{0(i,j)}, \mathcal{T}(c, x_0)_{(i,j)}]$ ($i, j$ are position indexes, and supposing $x_{0(i,j)} < \mathcal{T}(c, x_0)_{(i,j)}$), both loss terms are enabled, otherwise, only $\mathcal{L}_{\epsilon-pred}$ is enabled.

The complete training loss of Scene Diffusion is:

$$\mathcal{L} = \mathcal{L}_{\epsilon\_pred} + \lambda_{saca}\mathcal{L}_{saca} \tag{7}$$

$\lambda_{saca}$ is the manually set weight, in our practice, setting it to 1 gives satisfactory results.

## 3.3 Frequency Progression Training Schedule

In Scene Diffusion, the object and background areas are prompted by the condition image and the scene description respectively. Since the primary information about object is already provided by the condition image, it is highly probable that its denoising process will outpace the background area's. Concretely, the high-frequency signals in the object area may growing faster than in the others. Here we employ the indicator of Energy of Gradient (EoG) to support this reasoning, which is commonly used to quantify the content of high-frequency signals of image. It is calculated on the grayscale, formulated as:

$$EoG(x) = \frac{1}{hw}\Sigma_{i=1}^{h}\Sigma_{j=1}^{w}[(x_{(i+1,j)} - x_{(i,j)})^2 + (x_{(i,j+1)} - x_{(i,j)})^2] \tag{8}$$

A higher EoG indicates a more significant variation within a narrow range and a stronger presence of high-frequency signals. We calculate the EoG on object and background areas of $\hat{x}_0$ in each timestep, the trends are shown in Figure 3. As the background typically consists of sizeable regions of uniform color, its EoG reference value is lower than that of the object area. As shown in plots, the high-frequency signals of two areas do have the different growth curves. When no targeted strategy is applied (orange), the EoG of object area undergoes a rapid increase in the early stage. Despite the decline afterwards, its final value still exceeds the reference value, which indicates the disharmonious visual effect.

We propose the Frequency Progression Training Schedule (FPTS) to alleviate this problem. The core concept is applying SACA to different frequency bands of the image for different timesteps. When $t$ is large, the network mainly predicts the coarse information of

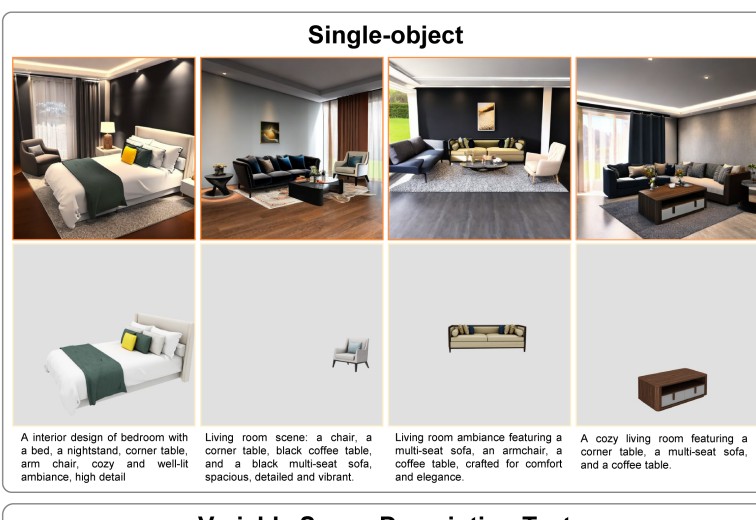

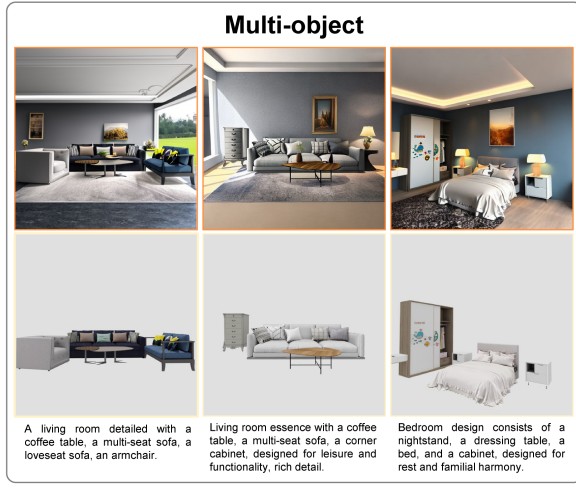

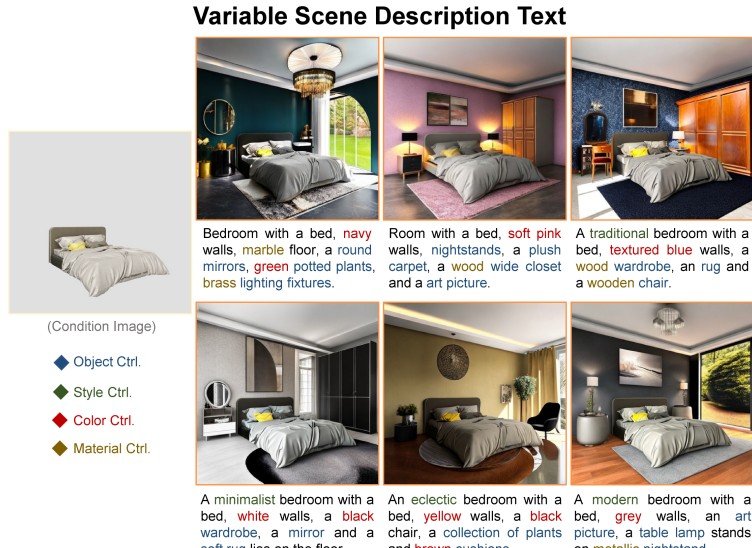

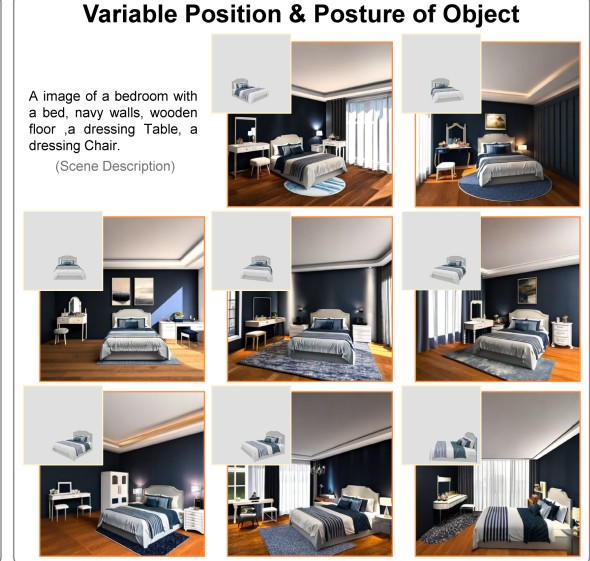

**Figure 4: The samples of generated scene images of Scene Diffusion. The corresponding condition image is placed at the bottom or upper left corner. To simplify, we omit the common prompt words about image quality in scene description.**

image, with SACA being applied solely on the low frequency bands. As $t$ decreases, the network begins to predict finer details, and the effective range of SACA will progressively extend to higher frequencies. The low-frequency signals of image are separated by standard gaussian downsample operation $\mathcal{G}$, which first convolves the image using a $3x3$ standard gaussian kernel and then applying $2x2$ average pooling.

In practice, $\mathcal{L}_{saca}$ is conducted between $\mathcal{G}(\hat{x}_0)$ and $\mathcal{G}(\mathcal{T}(c, x_0))$ when $700 > t \geq 300$, while between $\mathcal{G}^2(\hat{x}_0)$ and $\mathcal{G}^2(\mathcal{T}(c, x_0))$ when $t > 700$. Here $\mathcal{G}^2$ represents the two-stage cascaded $\mathcal{G}$ operation. The thresholds for $t$ are selected by experience. As shown in Figure 3, The application of FPTS (blue) softens the rise of high-frequency signals in the object area, and ultimately maintains it at a reasonable level. In the following section, we will further provide intuitive comparisons to validate its effectiveness on improving visual harmony of synthesized image.

## 4 EXPERIMENTS

The numerical experiments are performed to evaluate the proposed Scene Diffusion. This section starts by introducing the basic experimental setup. Afterwards, the results of the Scene Diffusion and its comparison with the other alternative methods are reported. Finally, the ablation study and expanded applications for proposed method are demonstrated.

### 4.1 Experimental Settings

**Dataset** We employ the public interior design dataset 3D-FUTURE [6, 7] to evaluate the proposed Scene Diffusion framework. It consists of 20240 clean and high-quality scene rendering images and 16563 textured 3D model of furniture. Each scene rendering image comes with complete annotations, including the fine categories and

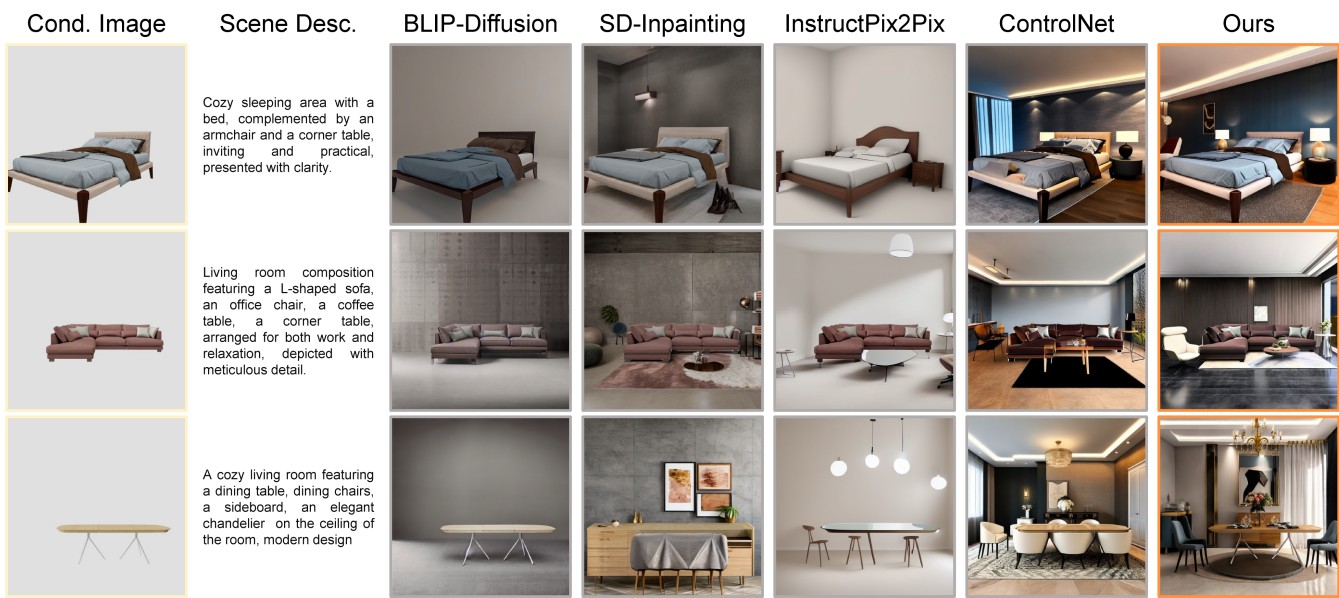

**Figure 5: The samples of generated scene images of the comparison methods. All compared methods are implemented with the recommended settings. The ControlNet was trained with our dataset, using the same hyperparameters as the proposed method.**

poses of included objects and camera setup, which allows us to construct the envisioned condition-text-output triple data.

We first split the dataset. We select 50 objects for each of the five major furniture categories (bed, sofa, table, chair and shelf) as the test data, and make sure that the scene images containing them are not used for training. Next is to get the description for each scene image. We initially construct the concise description comprised solely of nouns, such as "Bedroom design, a king-size bed, nightstands, an armchair". Then the multimodal language model LLaVA-1.5 [16] is employed to further polish it from aspects of overall style and object appearance. Finally, we render the single-object condition image under the given object pose and camera setup. This step is conducted based on Blender 3.6. Since a scene image usually contains multiple furniture, it will correspond to multiple condition images. Ultimately, there are 49,963 condition-text-output triples are used for training.

**Implementation Details** Scene Diffusion are trained based on Stable Diffusion V2.1, and its network architecture is exactly the same as ControlNet. The training is conducted on $512 \times 512$ resolution. AdamW [17] optimizer is used with a constant learning rate of $1e-5$. 4 Nvidia V100 are used in training phase, with a global batch size of 8 and a total iteration of 40,000. All the synthesized images shown in this article are sampled by DDIM [29] sampler, with the step size of 100 and the classifier-free guidance scale of 7.

**Evaluation Metrics** Four metrics are utilized in this section to fully assess the effect of the proposed method, including Image Entropy (**IE**), CLIP-based Text-Image Similarity (**C-TIS**) [23], Appearance Fidelity Score (**AF Score**) and Image Quality Score (**IQ Score**). **IE** is a widely accepted objective metric for assessing the amount of information in the image. The higher IE commonly signifies the more diverse composition within the generated image. It is calculated on the greyscale, formulated as: $IE(x) = -\Sigma_{i=1}^{255} p_i \cdot log_2 p_i$,

in which $p_i$ represents the probability that the gray value of image pixel is equal to $i$. **C-TIS** is defined as the cosine similarity between the CLIP features of the synthesized image and the text prompt, which reflects the level of model's controllability. **AF Score** and **IQ Score** are measured by user study. Fifty individuals take part in the survey and assess the results of our method and other comparisons in regards to the appearance fidelity of object and overall image quality. Our questionnaire explicitly emphasizes the importance of visual harmony in determining image quality. The scale goes from 1 to 5, with a higher score indicating better results.

## 4.2 Application Effect of Scene Diffusion

By manipulating the condition image and scene description, Scene Diffusion can accomplish a variety of control objectives. In this section, we mainly demonstrate its application effects in four aspects, the samples are shown in Figure 4. **1) Single-object.** Generating the complete scene image conditioning on the single-object rendering is the basic task for the proposed method. As can be seen, Scene Diffusion is capable of faithfully drawing the conditional object into the image while providing it a suitable shading that cohering with the global environment. This property functions well even when addressing small objects like chairs. **2) Multi-object.** We found that the network trained with single-object condition images generalizes well to the multi-object situations. It is our conjecture that the acquisition of this ability is related to the network's learning to the objects with complex structures. For the network, processing a combination of multiple objects is essentially the same as processing a single object with multiple different parts. This feature further increases the practicality of the proposed method in the design work. **3) Variable Scene Description Text.** Since the Scene Diffusion is trained based on the large-scale T2I model, the latter's strong semantic control ability can also be utilized in sampling

**Table 1: Quantitative Evaluation of Comparison Methods**

| Method | IE | C-TIS | AF Score | IQ Score |
|---|---|---|---|---|
| BLIPDiffusion [15] | 6.35 | 26.93 | $3.45_{\pm1.21}$ | $3.00_{\pm1.30}$ |
| SD-Inpainting [25] | 7.16 | 29.88 | $4.01_{\pm1.09}$ | $3.78_{\pm1.13}$ |
| InstructPix2Pix [2] | 7.11 | 30.19 | $2.66_{\pm1.35}$ | $2.97_{\pm1.22}$ |
| ControlNet [39] | 7.74 | 31.64 | $3.52_{\pm1.08}$ | $3.98_{\pm1.03}$ |
| Ours | **7.78** | **31.92** | $\mathbf{4.39_{\pm0.91}}$ | $\mathbf{4.48_{\pm0.88}}$ |

process. By modifying the scene description, we can manipulate the elements such as scene style and the occurrences, colors and materials of objects in the background. **4) Variable Position & Posture of Object.** As shown in the figure, when the position and posture of condition object shift, the network can still produce the image with reasonable layout and the required scene semantics. This feature facilitates the user to continuously adjust the object's pose to achieve more satisfying visual effect.

## 4.3 Comparison with Existing Alternatives

Several advanced alternative methods are selected for comparison, including BLIP-Diffusion [15] (**personalization -based**), SD-Inpainting [25] (**editing-based**), InstructPix2Pix [2] (**editing-based**), and ControlNet [39] (**learning-based**). Table 1 and Figure 5 provide the results of the quantitative and qualitative comparisons respectively. The metrics of IE and C-TIS are calculated with all 250 test cases, and AF Score and IQ Score are counted based on the assessments of 50 users on 30 test cases. Since the users may have diverse preferences, we additionally report the standard deviations of the last two metrics. Please refer to supplementary materials for the implementation details of compared methods.

According to the reported outcomes, Scene Diffusion surpasses existing alternatives in both of evaluation indicators and intuitive visual effects, which confirms its superiority on scene image synthesis task. Next, we will analysis the sources of its efficacy by contrasting it with different types of alternatives.

The personalization-based method does not achieve satisfactory results on this task. BLIP-Diffusion lags behind other methods in both IE and C-TIS indicators. The images produced by it generally lack a defined background. This result is derived from the basic principle of this type of methods. Personalization-based methods are initially designed to allow the network to generate the images about a specific object, which emphases on preserving the object's identity rather than controlling its poses. Although the combining with edge-driven ControlNet can partially serve the latter purpose, the edge map obtained from the single-object rendering image will in turn restrict the generation of background area. Overall, to meet the demands of this task, a unified control mechanism for object's appearance and pose is necessary.

Comparing to BLIP-Diffusion, the editing-based methods shown better ability in drawing objects in the background that approximately align with the text prompts. Both SD-Inpainting and InstructPix2Pix acquire the IE and C-TIS surpassing BLIP-Diffusion at least 0.76 and 2.95 respectively. However, these methods have their limitations when tackling this task, such as the changeless object colors in SD-Inpainting's results and unfaithful object appearances

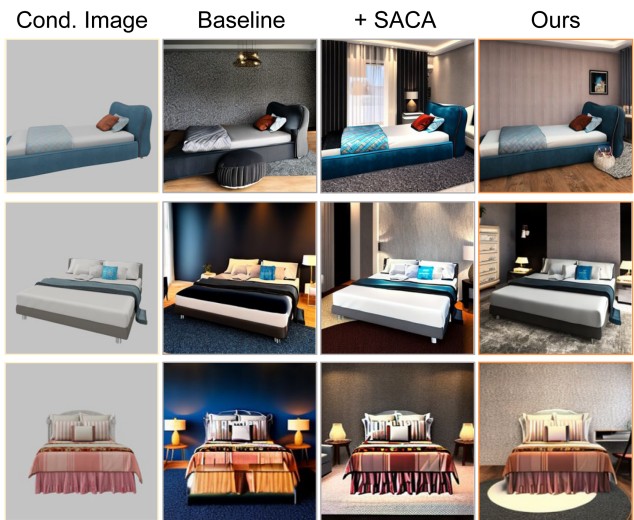

**Figure 6: The generated images under different ablation setups. The images in same row use the same scene description. The images are cropped to better showcase the object area. Please refer to supplementary materials for original images.**

**Table 2: Quantitative Evaluation for Ablation Study**

| Setup | IE | C-TIS | AF Score | IQ Score |
|---|---|---|---|---|
| Baseline | 7.77 | 30.22 | $3.38_{\pm1.17}$ | $3.63_{\pm1.20}$ |
| + SACA | **7.83** | 30.57 | $3.78_{\pm1.06}$ | $3.67_{\pm1.12}$ |
| Ours | 7.77 | **30.82** | $\mathbf{4.26_{\pm0.96}}$ | $\mathbf{4.07_{\pm1.02}}$ |

in InstructPix2Pix's results, and the latter leads to the lowest AF score among all the comparisons. The causes of these method- specific drawbacks have been discussed in section of related works. A noticeable phenomenon is that the editing-based methods may occasionally misinterpret the structure of objects. Take the third case in Figure 5 as the example, the table legs are mistakenly drawn as the part of chair back or the chair legs. This observation implies that the use of feature manipulation in editing-based methods may alter the sematics of small parts of complex objects.

The most critical comparison lies between ControlNet and the proposed method. As the learning-based method, their performance surpasses that of other methods in scene synthesis task. This leading position is also reflected in the quantitative indicators. ControlNet acquires the IE, C-TIS, and IQ Score higher than the third best method 0.58, 1.45 and 0.2 respectively, which confirms the advantages of learning-based method in image quality and controllability. The only exception occurs in AF Score. ControlNet's results do not receive high ratings in terms of appearance fidelity. This phenomenon can be attributed to that the network may not accurately learn the relationship between the condition and the output image. Take the first and third cases in Figure 5 as the examples, the network mistakenly interprets the condition input on bed foot area and table leg area as the guidance for the image structure, which results in the unfaithful appearances and unsatisfactory visual effects. To tackle the problem, an explicit constraint between the condition

**Integrating with Existing ControlNet**

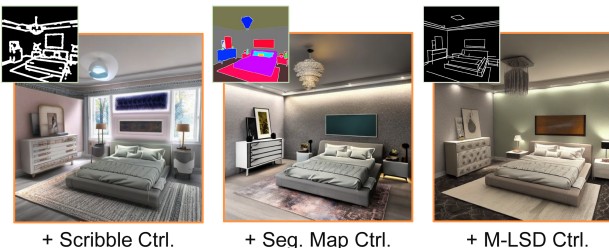

+ Scribble Ctrl.      + Seg. Map Ctrl.      + M-LSD Ctrl.

**Generalizing to Real Image Fragment**

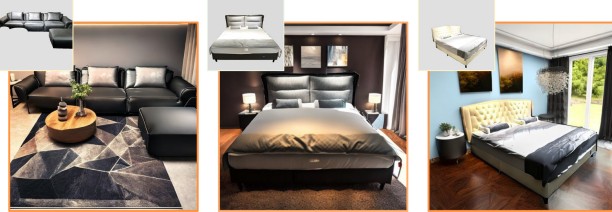

Good Case                                    Failure Case

**Figure 7: The results of expanded applications of Scene Diffusion. The condition images for integrated ControlNet or the fragments of real images are placed in upper left corner.**

and output image is required. In Scene Diffusion, this issue is fixed by the introduction of SACA. The inclusion of FPTS afterwards further enhanced the image quality from the perspective of visual harmony, which ultimately makes the Scene Diffusion achieve the best results among all the comparisons.

### 4.4 Ablation Study

Ablation study is conducted to evaluate the effects of proposed SACA and FPTS components. Since FPTS cannot be applied independently, in this part, we mainly compare three setups: **1) Baseline.** No improving component is applied, and the network is trained solely with $\mathcal{L}_{\epsilon-pred}$. **2) + SACA.** Only SACA is applied. **3) Ours.** Both SACA and FPTS are applied. All the networks in ablation study are trained with the subset of training data that using bed as the condition object. The quantitative and qualitative results are reported as Table 2 and Figure 6. IE and C-TIS are calculated with 50 test cases, and AF Score and IQ Score are counted based on the assessments of 50 users on 10 test cases.

As can be seen, the proposed components indeed bring improvements from the perspective of appearance fidelity of object and image quality. The setups with SACA yield the AF Score that is at least 0.4 higher than the baseline. However, using SACA in isolation also leads to adverse effects. As shown in Figure 6, the results of second setup exhibits a more "saturated" visual effect in the object area. In fact, this is a direct manifestation of the excessive high-frequency signals in this area. Excessive high-frequency signals also contribute to its IE indicator. Such phenomenon reduces the users' assessments in both image quality and appearance fidelity. This issue was eventually alleviated by FPTS. The third setup surpass the second setup 0.48 and 0.4 in AF Score and IQ Score respectively.

### 4.5 Expanded Applications

We investigate two key expended applications of Scene Diffusion, including integrating with existing ControlNet and generalizing to real image fragment. The results are shown in Figure 7.

**Integrating with Existing ControlNet** The proposed method controls the scene semantics through the text prompt. In some cases, the user may have more detailed requirements for the structure of scene. The integration of Scene Diffusion and the ControlNet can meet this demand. The results in Figure 7 confirms that Scene Diffusion is highly compatible with the structure-driven ControlNet that using scribble[30], segmentation map[43], or M-LSD [9] image as the condition input. To accomplish this expended application, we additionally train a network based on Sable Diffusion V1.5 to align with existing M-LSD-driven ControlNet.

**Generalizing to Real Image Fragment** Except for the single-object rendering image obtained from 3D software, the object fragment from real image may also serve as the condition image of the proposed method. To investigate its effect on this important application, we collect some in-the-wild images of furniture and conduct the test. As shown in Figure 7, our approach shows a certain degree of generalization ability on real image fragments. For first two test cases, it can synthesize the image with faithful appearance of object and harmony visual effect. However, for the third test case, the network does not produce satisfactory result. We address this issue to the complex lighting environment in orginal image. The strong diffuse reflection and specular highlight impede the network's perception to the color of object. Due to the limited capability of 3D software in simulating real lighting, one feasible solution to this challenge is introducing real 2D images into the training phase, which is also a topic that we plan to explore in future research.

## 5 CONCLUSION

In this work, we explore a novel task that synthesizing the scene image conditioning on the single 3D model and the scene description. After identifying the main goals of ensuring appearance fidelity of drawn object and visual harmony of entire image, we propose Scene Diffusion framework to meet the challenge. Comparing to the existing learning-based methods, the key novelty is introducing the SACA as an intensive training objective, which successfully promotes the appearance consistency between the condition and output image without hindering the network's learning to the prior of global shading coherence. Afterwards, FPTS is utilized to improve the visual harmony of entire image by moderating the growth of high-frequency signals in the object area. In contrast to traditional 3D-based pipeline, this framework eliminates the laborious scene construction step and offers enhanced adaptability in time-sensitive situations.

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
