# OpenReview forum: "Scene Diffusion: Text-driven Scene Image Synthesis Conditioning on a Single 3D Model"
_acmmm.org/ACMMM/2024/Conference — MM2024 Oral_

### Official Review · Reviewer_uH34 · 2024-05-08

**Rating:** 5
**Confidence:** 3

**Summary:**

In this work, authors propose Scene Diffusion to meet the challenges of  appearance fidelity and visual harmony.
The main contributions of this article are as follows:
1.Investigating a novel challenge of generating scene imagesbased on a single 3D model with the prior of the pre-trained T2I model. In contrast to conventional 3D-based pipeline, this method streamlines the complex scene crafting process and is better suited for the scenarios that demand rapid evaluation.
2.Proposing a novel learning-based scene image synthesis framework. By introducing Shading Adaptive Condition Alignment as an intensive training objective, the framework effectively facilitates the network’s learning to the complex relationship between the condition and output image. The subsequent addition of Frequency Progression Training Schedule further improves the visual harmony of the synthesized image.
3.The proposed Scene Diffusion has been shown to excel beyond the other alternatives in terms of faithfully presenting condition object and generating high quality images. In addition,it also shows the ability to seamlessly integrate with existing ControlNet and potential for generalizing to real image fragments.

**Strengths:**

1. The paper has a clear and logical structure and is well written.
2. The results are clear and reliable. It can be seen from the experiment that the author has fully visualized and comparatively analyzed the results.
3. The author proposes FPTS to optimize the results of EOG and adjust the harmony of the background and objects, which is innovative and has obvious optimization effects.

**Limitations:**

1. Although the visualization effects in the paper all reflect the harmonious background and inter-object effects, the shadow effects of light in some pictures still need to be improved. If some failure cases can be shown to further verify some infeasible effects, it may lead to new directions of inquiry for future work.
2. The examples used in the experiment seem a bit insufficient. There are only a few types of furniture such as beds, sofas, shelves, etc. I would like to know if there are more examples.
3. I am curious about how long the whole process will take, and whether there are other papers that have introduced EOG analysis on images. This will be of great help to me in judging the innovativeness of the paper.

**Suitability:**

2

---

### Official Review · Reviewer_HjEi · 2024-05-23

**Rating:** 4
**Confidence:** 3

**Summary:**

This paper introduces Scene Diffusion generating scene images for product design evaluation. Key challenges addressed include maintaining object appearance fidelity and overall visual harmony. The proposed framework includes Shading Adaptive Condition Alignmen  for appearance consistency and a low-to-high Frequency Progression Training Schedule for visual coherence.

**Strengths:**

1. The paper addresses an important topic in scene generation, focusing on specific 3D objects, which is highly relevant for product design and evaluation.
2. The demonstrated examples in the paper exhibit high visual quality, showcasing the effectiveness of the proposed method in generating realistic scene images.

**Limitations:**

1. The paper lacks clear demonstration of improvements over inpainting-based methods, making it difficult to assess the effectiveness of the proposed approach compared to existing techniques.
2. Certain highlighted abilities such as style color material control appear disconnected from the core contributions of the method, potentially detracting from clarity regarding its primary benefits.
3. Quantitative evaluation metrics show minimal differentiation and high variance, raising concerns about the reliability of quantitative assessments for image generation tasks, though this is a common challenge in the field.

**Suitability:**

3

---

### Official Review · Reviewer_Dq8q · 2024-05-26

**Rating:** 4
**Confidence:** 3

**Summary:**

The proposed work presents a novel conditional synthesis method to create the scene image based on a single 3D model of the desired object and description of the scene. A major challenge is maintaining the object's fidelity while maintaining overall consistency with the text description.

**Strengths:**

- **Shaping Adaptive Condition Alignment**: They propose a novel framework that maps complex relationships between the condition and the output image.
- **Exhaustive Comparison**: The proposed method shows superior performance when compared with other baseline methods like SDXL-inpainting, ControlNet etc.

**Limitations:**

- **Conditioning with multiple objects**: Can this method also work when two objects are given as input for conditioning?
- **Consistency of Generation**: The proposed method uses a rendered image of a 3D model as a condition image. Does the proposed framework generate a similar-looking scene when the pose of the object is changed, but the text description is the same? How much will this variation be? Is this a drawback of this method, or is it expected?

**Suitability:**

3

---

### Official Review · Reviewer_cCSR · 2024-05-29

**Rating:** 5
**Confidence:** 3

**Summary:**

The task addressed in this article is: given a 2D rendered image of a 3D object and a scene description, generate a 2D scene image that includes the 2D object and matches the description. There are two issues: first, the shading in the 3D object’s rendered image may not match the shading described in the text prompt. To unify the shading, they normalize the distribution of the object area during training using the mean and standard deviation of the ground truth image $x_0$. Second, to align the denoising process of the object and background areas, they apply SACA (Shading Adaptive Condition Alignment) to the low-frequency band at large timesteps and to the high-frequency band at small timesteps.

Overall, the method is simple but effective, and experiments have demonstrated its effectiveness. My main concern is whether this task is novel. Authors should provide sufficient comparisons with related tasks; otherwise, I would lower my rating.

**Strengths:**

1. The method introduces a shading adaptive transformation to ensure that the lighting of the rendered 3D object image matches the lighting described in the text description, addressing potential discrepancies.
2. Adding a Frequency Progression Training Schedule (FPTS) further improves the visual harmony of the synthesized image.

**Limitations:**

1. The paper claims to propose a completely new task, generating a scene image based on a 3D model and scene description. However, it does not provide references to demonstrate the importance of this task, nor does it explain how it differs from other editing tasks. Therefore, the motivation of the entire paper is not convincing.
2. Section 3.1 spends a lot of space describing the stable diffusion model, but this is not the authors’ contribution.
3. The writing quality is subpar, making it difficult to get the point of SACA, and many equations contain errors and lack punctuation.
4. Will FPTS be used during inference?

**Suitability:**

2

---

### Meta-Review · Area_Chair_P7S7 · 2024-07-01

**Recommendation:** Accept (Oral)
**Confidence:** 4

**Metareview:**

All reviewers favor to accept the paper. The rebuttal has cleared most concerns. The AC agrees with the reviewers and recommends acceptance.